# Preferential Expression of Programmed Death Ligand 1 Protein in Tumor-Associated Macrophages and Its Potential Role in Immunotherapy for Hepatocellular Carcinoma

**DOI:** 10.3390/ijms22094710

**Published:** 2021-04-29

**Authors:** Dong-Jun Park, Pil-Soo Sung, Gil-Won Lee, Sung-Woo Cho, Sung-Min Kim, Byung-Yoon Kang, Won-Hee Hur, Hyun Yang, Soon-Kyu Lee, Sung-Hak Lee, Eun-Sun Jung, Chang-Ho Seo, Joseph Ahn, Ho-Joong Choi, Young-Kyoung You, Jeong-Won Jang, Si-Hyun Bae, Jong-Young Choi, Seung-Kew Yoon

**Affiliations:** 1The Catholic University Liver Research Center, Department of Biomedicine & Health Sciences, College of Medicine, The Catholic University of Korea, Seoul 06591, Korea; akhpdyj@catholic.ac.kr (D.-J.P.); pssung@catholic.ac.kr (P.-S.S.); lgw0429@catholic.ac.kr (G.-W.L.); 1102csw@catholic.ac.kr (S.-W.C.); wildtultul@catholic.ac.kr (S.-M.K.); kby2132@catholic.ac.kr (B.-Y.K.); wendyhur@catholic.ac.kr (W.-H.H.); oneggu@catholic.ac.kr (H.Y.); blackiqq@catholic.ac.kr (S.-K.L.); garden@catholic.ac.kr (J.-W.J.); baesh@catholic.ac.kr (S.-H.B.); jychoi@catholic.ac.kr (J.-Y.C.); 2Department of Internal Medicine, College of Medicine, Seoul St. Mary’s Hospital, The Catholic University of Korea, Seoul 06591, Korea; 3Department of Internal Medicine, College of Medicine, Eunpyeong St. Mary’s Hospital, The Catholic University of Korea, Seoul 03383, Korea; 4Department of Clinical Pathology, Seoul St. Mary’s Hospital, College of Medicine, The Catholic University of Korea, Seoul 06591, Korea; hakjjang@catholic.ac.kr; 5Department of Hospital Pathology, College of Medicine, Eunpyeong St. Mary’s Hospital, The Catholic University of Korea, Seoul 03383, Korea; esjung@catholic.ac.kr; 6Department of Surgery, College of Medicine, Seoul St. Mary’s Hospital, The Catholic University of Korea, Seoul 06591, Korea; schjee@catholic.ac.kr (C.-H.S.); 21000617@cmcnu.or.kr (J.A.); hopej0126@catholic.ac.kr (H.-J.C.); yky602@catholic.ac.kr (Y.-K.Y.)

**Keywords:** hepatocellular carcinoma, tumor-associated macrophages, PD-L1, anti-PD-L1

## Abstract

A predictive biomarker of immune checkpoint inhibitor (ICI)-based treatments in hepatocellular carcinoma (HCC) has not been clearly demonstrated. In this study, we focused on the infiltration and programmed death ligand 1 (PD-L1) expression of tumor-associated macrophages (TAMs) in the tumor microenvironment of HCC. Immunohistochemistry demonstrated that PD-L1 was preferentially expressed on CD68^+^ macrophages in the tumor microenvironment of HCC, suggestive of its expression in TAMs rather than in T cells or tumor cells (*P* < 0.05). A co-culture experiment using activated T cells and M2 macrophages confirmed a significant increase in T cell functionality after the pretreatment of M2 macrophages with anti-PD-L1. Syngeneic mouse model experiments demonstrated that TAMs expressed PD-L1 and tumors treated with anti-PD-L1 showed smaller diameters than those treated with IgG. In these mice, anti-PD-L1 treatment increased activation markers in intratumoral CD8^+^ T cells and reduced the size of the TAM population. Regarding nivolumab-treated patients, three of eight patients responded to the anti-PD-1 treatment. The percentage of Ki-67-positive CD4^+^ and CD8^+^ T cells was higher in responders than non-responders after nivolumab. Overall, PD-L1 expression on TAMs may be targeted by immune-based HCC treatment, and ICI treatment results in the reinvigoration of exhausted CD8^+^ T cells in HCC.

## 1. Introduction

Hepatocellular carcinoma (HCC) accounts for approximately 80% of primary liver cancer, which is the sixth most common type of malignant tumor and the third major cause of cancer-related deaths globally [1,2]. The incidence of HCC has increased by 80% worldwide over the last 20 years [3]. At present, sorafenib and lenvatinib, multi-kinase inhibitors (MKIs), are used as the first-line oral therapy for unresectable HCC [4], although the survival benefits from these drugs are modest.

Immune checkpoint inhibitors (ICIs) have been reported to prolong the overall survival of patients with different types of malignancies. However, the tumor response rate of nivolumab monotherapy, the first approved anti-programmed death (PD)-1 monoclonal antibody, is below 20% in HCC [5]. A definite correlation between the expression of programmed death ligand 1 (PD-L1) in tumor cells and responses to anti-PD-1 has not been clearly demonstrated in HCC, unlike other solid tumors [6]. Immune heterogeneity has been observed in HCC and is known to contribute to ICI resistance [7,8]. To overcome this heterogeneity, clinical trials using combination therapy comprising ICI and MKI were performed and have demonstrated synergistic antitumor activity [9].

Currently, in daily clinical practice, there are no biomarkers that can reliably predict the response to immune-based therapy in HCC [10]. In general, tumor-specific CD8^+^ T cells play an important role in killing cancer cells. The total number and distribution of CD8^+^ T cells in tumors are known to influence tumor development and responsiveness to cancer immunotherapy [11,12]. However, the tumor microenvironment (TME) is composed of various types of immune cells. In particular, the infiltration of several types of immune cells, such as tumor-associated macrophages (TAMs), was reported to induce immune suppression in the TME of HCC [3]. Typically, macrophages can polarize into classically activated macrophages (M1-type) and alternatively activated macrophages (M2-type) in response to various microenvironmental signals [13]. In general, TAMs exhibit the M2 phenotype, and their frequency is known to be associated with poor prognosis of HCC [14,15,16], although the function and phenotypes of TAMs in human HCC tissues have not been clearly characterized. PD-L1 in tumor cells may directly suppress exhausted PD-1^+^ T cells. However, the importance of PD-L1 expressed in TAMs remains unclear [17]. In this study, we focused on the infiltration and PD-L1 expression of TAMs in the TME of HCC.

## 2. Results

### 2.1. Infiltration of CD3^+^ T Cells and CD68^+^ Macrophages in the HCC Tissue of Patients

To confirm the expression of PD-L1 in CD68^+^ macrophages from HCC patients, we performed immunohistochemistry for liver biopsy or surgically removed tissue samples from 33 patients with HCC. The results showed that the number of CD3^+^ T cells, CD68^+^ macrophages, and PD-L1^+^ cells was higher in the peritumoral region than in the intratumoral region (*P* < 0.001) (Figure 1A–E). In particular, CD3^+^ T cells and CD68^+^ macrophages were confirmed to be distributed in different patterns, and PD-L1 was expressed in a similar pattern to CD68 (Figure 1A). Moreover, the number of PD-L1-expressing cells positively correlated with the number of CD68^+^ macrophages (Figure 1D middle), but not with the number of CD3^+^ T cells (Figure 1D left). The number of PD-L1^+^ cells in the intratumoral region showed no significant correlation with the number of CD68^+^ macrophages (Figure 1E middle). In contrast to the peritumor regions, the number of CD3^+^ T cells and the number of PD-L1^+^ cells were positively correlated in the intratumor regions (Figure 1E, left). Lastly, we compared the number of PD-L1^+^ cells in the peritumor and intratumor regions with the concentration of serum alpha fetoprotein (AFP) and confirmed that there was no correlation (Figure 1D, right and Figure 1E, right).

### 2.2. Improvement in CD8^+^ and CD4^+^ T Cell Functions after PD-L1 Expression Blockade on M2 Macrophages

Next, we studied whether CD8^+^ and CD4^+^ T cell functions are induced upon the blockade of PD-L1 expression on M2 macrophages. We isolated PBMCs from healthy donor blood and stained them with CD14 and CD3 microbeads for magnetic cell sorting. CD14^+^ cells were then polarized into M2 macrophages through treatment with M-CSF and IL-4. After polarization, CD3^+^ T cell co-culture experiments were performed. In co-cultures, we observed functional enhancements of the CD8^+^ T cells co-cultured with PD-L1-pretreated M2 macrophages. The numbers of CD8^+^ IFN-γ^+^ T and CD8^+^ TNF-α^+^ T cells significantly increased by 5% to 10% and 8% to 10%, respectively (Figure 2A,B). Moreover, PD-1 and CD69 expression significantly increased on PMA/Ionomycin-activated CD8^+^ T cells after PD-L1 blockade on M2 macrophages (Appendix A). Consistent with the observations reported for CD8^+^ T cells, PMA/Ionomycin-activated CD4^+^ INF-γ^+^ T cells increased by approximately 8% to 14%, while the CD4^+^ TNF-α^+^ T cell population increased by approximately 7% to 9% (Figure 2C,D). Further, CD4^+^ T cells showed an increase in the expression of PD-1 and CD69 after PD-L1 expression blockade on M2 macrophages (Appendix A).

Next, we performed similar experiments using CD206^+^ macrophages and CD3^+^ T cells freshly isolated from the HCC tissue of a patient. PD-L1 expression was blocked on CD206^+^ macrophages using anti-PD-L1 antibody, and then these cells were co-cultured with PMA/Ionomycin-activated CD3^+^ T cells (Figure 2E). We found that the population of CD8^+^ INF-γ^+^ T cells increased by approximately 8% to 13%, while that of CD4^+^ INF-γ^+^ T cells increased by approximately 9% to 12% (Figure 2F). Further, CD8^+^ TNF-α^+^ T cell population increased by approximately 13% to 16% and CD4^+^ TNF-α^+^ T cells increased by approximately 11% to 14% (Figure 2G).

### 2.3. Effects of Anti-PD-L1 Treatment in a Syngeneic HCC Mouse Model

To confirm the in vitro experimental findings, we performed in vivo experiments using a syngeneic HCC mouse model. First, we demonstrated the immunogenicity of the tumor derived from Hepa1-6 cells (mouse liver cancer cells), allowing us to generate an adequate experimental model [18,19]. Both CD8^+^ T cells and CD11b^+^ F4/80^+^ macrophages were acquired from both the tumor and spleen, simultaneously. The expression of PD-L1 on CD11b^+^ F4/80^+^ macrophages and PD-1 in CD8^+^ T cells was analyzed using flow cytometry (Appendix A). PD-L1 expression was much higher on CD11b^+^ F4/80^+^ TAMs than on spleen macrophages and hepa1-6-derived tumor cells. Moreover, expression of PD-L1 was increased in IFN-γ-stimulated Hepa1-6 cells (Figure 3A). After blocking PD-L1 using the anti-PD-L1 antibody, the PD-L1 FACS antibody hardly detected the PD-L1 molecule in Hepa1-6 cells (Appendix A). In CD8^+^ T cells, the mean fluorescence intensity (MFI) of PD-1 and CD11b^+^ F4/80^+^ macrophages in the tumor was significantly higher than the MFI of PD-L1 in the spleen (Figure 3B). To block PD-L1 expression, anti-PD-L1 was intraperitoneally injected into the mice on day 3, 8, and 14 after Hepa1-6 cell injection (Figure 3C). The tumor size rapidly increased in the mock-treated group as compared to that in the group treated with anti-PD-L1 (Figure 3D). CD8^+^ T cells were isolated from excised tumors and their activation was identified through CD69 expression evaluation. We verified a significant increase in the MFI of CD69 in the group injected with anti-PD-L1 (Figure 3E). Furthermore, the number of CD11b^+^ F4/80^+^ macrophages per 1g tumor weight was significantly reduced in the group injected with anti-PD-L1 (Figure 3F).

### 2.4. Ki-67 Expression in Bulk CD8^+^ and CD4^+^ T Cells from Patients Reflects Nivolumab Response in HCC

Finally, we analyzed peripheral blood samples of the HCC patients treated with nivolumab. We divided these patients into a complete response (CR) + partial response (PR) group (*n* = 3) and stable disease (SD) + progressive disease (PD) group (*n* = 5). PBMCs were obtained from patients from each group at two time points: before nivolumab administration and after nivolumab administration for 4 weeks. CD8^+^ T cells from CR or PR patients showed approximately 2% Ki-67 expression before nivolumab administration, which increased to around 4% after 4 weeks (Figure 4A). Furthermore, Ki-67 expression in CD4^+^ T cells increased from around 3% to 4% after administration of nivolumab (Figure 4B). However, in SD or PD patients, no significant change was observed in Ki-67 expression level before and after nivolumab administration. The CR or PR group showed higher expression of Ki-67 in both CD8^+^ and CD4^+^ T cells than the SD or PD group. Next, we were unable to obtain tissues from all nivolumab-treated patients owing to the limited sample availability. We performed immunohistochemistry using the tissues from one CR patient and one PD patient and used them as the representative data. We confirmed the distribution of cells expressing CD3, CD68, and PD-L1. Based on this assay, we identified that CD3^+^ T cells and CD68^+^ macrophages were more abundantly distributed in the peritumoral region than in the intratumoral region (Figure 4C,D). Interestingly, CD68^+^ macrophages in the peritumoral region of the tumor with CR robustly expressed PD-L1 (Figure 4C). On the other hand, numerous CD68^+^ macrophages were observed both intra- and around the tumor of the patient with PD, but no PD-L1 expression was detected (Figure 4D).

## 3. Discussion

In this study, we demonstrated the expression of the PD-L1 protein on TAMs and its potential predictive role in immunotherapy for HCC. Using the tissues of HCC patients, we verified PD-L1 expression on TAMs in the TME. In addition, we confirmed for the first time the expression of Ki-67 in CD8^+^ and CD4^+^ T cells from PBMCs of patients with HCC using nivolumab. We demonstrated that the function of CD8^+^ and CD4^+^ T cells was restored by inhibiting the expression of PD-L1 on macrophages. Our results suggest that PD-L1-expressing TAMs may be useful as indicators of cancer immunotherapy in HCC.

ICIs interfere with the PD-1/PD-L1 interaction and are effective against several progressive cancers [20]. In the case of unresectable HCC, the Checkmate-040 study compared the objective response (OR) after using both nivolumab and a placebo. The results showed that the OR for nivolumab was 27% when PD-L1 was expressed in more than 1% of tumor cells. However, when PD-L1 expression in tumor cells was less than 1%, OR for nivolumab was 17% [6]. In the aforementioned study, the association between the frequency of total intratumoral CD68^+^ macrophages and OS was not significant [21]. In addition, another recent study demonstrated that specific genetic patterns or tumor mutational burdens were not related to treatment response [22].

Different types of CD4^+^ T cells in tumor microenvironments may also contribute to the tumor progression. Previous data from Li et al. [23] demonstrated that PD-1 is expressed in Th-1 cells in tumor microenvironments and that anti-PD-1 treatment reinvigorated the antitumor activity of these cells. Therefore, treatment with anti-PD-L1 could enhance the anticancer effect by improving the functionality of Th-1 cells [23]. Kamada et al. demonstrated that some gastric cancer patients show tumor progression despite ICI treatment. The reason is that PD-1^+^ FoxP3^high^ CD45RA^−^ CD4^+^ T (effector Treg) cells infiltrate the tumor and inhibit effector T cell activity. In such cases, treatment with anti-PD-L1 can induce tumor progression due to the increased activity of effector Treg cells [24].

In general, macrophages can be polarized into the M1 type (classically activated) or M2 type (alternatively activated). M1-type macrophages are derived upon stimulation with granulocyte–macrophage colony-stimulating factor (GM-CSF) alone or in combination with stimulated lipopolysaccharides (LPS) and IFN-γ. These cells exhibit antitumorigenic and pro-inflammatory functions. Many T cells or NK cells exist in the peritumoral region. PD-L1 expression on macrophages could be further increased by IFN-γ secreted by these cells [25]. On the other hand, the M2-type macrophages are induced after stimulation with M-CSF and IL-4. These cells have pro-tumorigenic and anti-inflammatory properties [26]. In HCC, TAMs are known to have characteristics similar to those of the M2-type macrophages [13,27]. In general, TAMs have significant immunosuppressive effects and, hence, are known to act on tumor angiogenesis, invasion, growth, and metastasis [27,28,29]. However, detailed research is lacking on the exact subtypes and functions of these TAMs in HCC [30,31]. A recent study has shown a decrease in TAM after anti-PD-L1 treatment in a mouse lung cancer model. Although the exact mechanism remains unclear, it is assumed that PD-L1-expressing macrophages treated with anti-PD-L1 may induce antibody-dependent cellular cytotoxicity (ADCC) by NK cells [32].

It seems that PD-L1-expressing tumor cells directly inhibit cytotoxic T cells, but the significance of PD-L1 expression on TAMs for the regulation of tumor-specific T cells remains unclear [17]. Until recently, there have been several reports demonstrating that PD-L1 is preferentially expressed on macrophages rather than cancer cells and may play a predictive role in immunotherapy. In oral squamous cell carcinoma, PD-L1 on CD163^+^ CD206^+^ TAMs directly inhibited cytotoxic T cells [33]. In another recent study, IHC double-staining was performed with PD-L1 and CD68 to indicate that PD-L1 expression is limited mainly to CD68-positive macrophages in HCC [34]. It is known that PD-L1 expression is generally very weak in HCC cells, and the expression of PD-L1 in TAMs may reflect tumor immunogenicity in the TME of HCC. In our study, the PD-L1-expressing TAMs in our HCC specimens did not appear to be fully M2-polarized, and they expressed high levels of HLA-DR (data not shown), suggesting that the tumor cells were immunogenic and might respond to treatments modulating antitumor immune responses [34,35].

## 4. Materials and Methods

### 4.1. Patient Samples and Clinical Information

This study was conducted under the Helsinki Declaration with the approval of the Seoul St. Mary’s Hospital Institutional Review Board of committee (KC19OESI0393, Approval date. 2019.07.08). From January 2019 to December 2020, the medical records of 33 patients who received liver biopsy or surgical intervention from our institution were reviewed. All patients were pathologically confirmed to have HCC after liver biopsy or surgery.

### 4.2. Flow Cytometry

Multicolor flow cytometry was performed using the following commercially available antibodies: phycoerythrin (PE)-conjugated anti-human PD-1, allophycocyanin (APC)-cyanine 7 (Cy7)-conjugated anti-mouse F4/80 and CD4 (BioLegend, San Diego, CA, USA), PE-conjugated anti-mouse PD-1, PerCP-Cy5.5-conjugated anti-mouse CD8a (eBioscience, San Diego, CA, USA), V500-conjugated anti-human CD3, V450-conjugated anti-human CD8, APC-H7-conjugated anti-human CD4, APC-conjugated anti-human CD69, V450-conjugated anti-mouse CD3 and CD11b, APC-conjugated anti-mouse CD69, V500-conjugated anti-mouse CD45, APC-conjugated anti PD-L1, and PE-Cy7-conjugated anti-mouse Ly6G (BD Biosciences, San Jose, CA, USA). Dead cells were excluded using the aqua fluorescent LIVE/DEAD dye (Invitrogen). Multicolor flow cytometry was performed using the LSR Fortessa, Canto II instrument (BD Biosciences). Data were analyzed using FlowJo software (TreeStar, Ashland, OR, USA).

### 4.3. Intracellular Cytokine Staining

Intracellular cytokine staining was performed as previously described [2,36]. Stimulated CD3^+^ T cells were treated with brefeldin A (5 μg/mL) (BD Biosciences) and monensin (5 μg/mL) (BD Biosciences). Next, cells were stained for surface markers after 7 h of incubation. Foxp3 Staining Buffer Kit (eBioscience) was used to permeabilize surface marker-stained cells and they were further incubated with PE-Cy7 anti-human Ki-67, tumor necrosis factor (TNF)-α, and PE anti-human interferon (IFN)-γ (BD Biosciences). Information on antibodies and proteins we used in this study can be found in Appendix A.

### 4.4. Purification of CD3^+^ T Cells and CD14^+^ Monocytes

Peripheral blood mononuclear cells (PBMCs) were isolated from a healthy adult donor using Ficoll-Hypaque density gradient centrifugation, as previously described [2]. CD3^+^ T cells (anti-CD3 microbeads, 130-050-101, MACS) and CD14^+^ monocytes (anti-CD14 microbeads, 130-050-201, MACS) were separated from PBMCs using the OctoMACS separator and starting kits (Miltenyi Biotec, Auburn, CA, USA).

### 4.5. Separation of CD206^+^ Macrophages in Human HCC Tissue

A dissociation kit (MACS, 130-095-929) was used to mash tumor tissue. Cells were stained with the APC-conjugated CD206 antibody in the dissociated solution. CD206^+^ macrophages (anti-APC microbeads, 130-090-855, MACS) were separated using OctoMACS separator and starting kits (Miltenyi Biotec).

### 4.6. Monocyte Polarization to M2 Macrophages

For M2 macrophage polarization, isolated human CD14^+^ monocytes were plated in 90 × 20 mm cell culture dishes (SPL Life Science, Korea) at a density of 1 × 10^6^ cells/mL in 10% Roswell Park Memorial Institute (RPMI)-1640 medium. Macrophage colony-stimulating factor (M-CSF; 1μg/mL) was added to fresh medium every other day. On day 6, the medium was refreshed with interleukin (IL)-4 (1μg/mL) and M-CSF (1μg/mL) for 24 h. Cells were maintained at 37 °C and 5% CO_2_ [37].

### 4.7. Co-Culture of CD3^+^ T Cells and M2 Macrophages

CD3^+^ T cells were pre-incubated for 6 h in serum-free RPMI-1640 medium for additional Ionomycin (1 μg/mL) and PMA (10 ng/mL). PMA/Ionomycin-activated CD3^+^ T cells (5 × 10^5^ cells/mL) were co-cultured with M2 macrophages (5 × 10^5^ cells/mL) for 24 h with mock or anti-PD-L1 (20 μg/mL) in RPMI-1640 medium containing 10% fetal bovine serum at 37 °C and 5% CO_2._

### 4.8. Immunohistochemistry

A 5-µm-thick cross-section of a paraffin-embedded block was moved to a silanized glass slide. The sections were then de-paraffinized using xylene and rehydrated using a series of graded alcohols. Antigen retrieval was performed using a microwave vacuum histoprocessor (RHS-1; Milestone, Bergamo, Italy) by heating samples in 0.01 M citrate buffer (pH 6.0) for 20 min to a final temperature of 121 °C. The section was incubated for 10 min with hydrogen peroxide (3%) in methanol to prevent endogenous peroxide activity. Slides were then incubated with anti-CD3 (Abcam), anti-CD68 (clone: KP1, Dako, Carpinteria, CA, USA), and anti-PD-L1 (clone: 22C3, Dako) antibodies. After washing, the EnVision+ system HRP-labelled polymer (Dako) was used at 24 °C for 5 min. The slides were treated with 3,3′-diaminobenzidine for 5 min and then counterstained with hematoxylin.

### 4.9. In Vivo Mouse Model

An in vivo mouse model was established as previously described [2]. Syngeneic mouse models of HCC were constructed by injecting 1 × 10^7^ Hepa1-6 cells into the flank of 6-week-old C57BL/6N mice. Next, anti-PD-L1 (Bio X Cell, NH, USA) antibody (100 μg) was injected as per the determined schedule. The size of the tumor was measured using a digital caliper and calculated using the formula DW^2^/2, where D is the depth and W is the width. A scalpel was used to remove tumor tissue and spleen from the mouse, which was then digested by collagenase (0.05%)/Hyaluronidase (1000 U/mL) and DNase (5 U/mL) to obtain a cell suspension. Subsequently, the supernatant was removed, and the pellet was treated with an RBC lysis buffer. The mixture was incubated at 24 °C for 5 min, and then centrifuged at 500× *g* for 5 min. After washing, supernatant was removed and the pellet was suspended in 100 µL of 1× phosphate-buffered saline (PBS) for fluorescence staining (CD3, CD4, CD8a, CD69, CD11b, F4/80, CD45, Ly6G, PD-1, PD-L1, and LIVE/DEAD dye) and FACS analysis (Approval no. CUMS-2018-0281-01, Approval date. 2018.10.04).

### 4.10. Statistical Analyses

GraphPad Prism version 7 software was used for all statistical analyses. The independent *t*-test was used for continuous variables. Pearson correlation tests were performed to analyze the correlation between the two parameters. Statistical significance was defined as * *P* < 0.05, ** *P* < 0.01, *** *P* < 0.001.

## 5. Conclusions

In this study, we have confirmed that PD-L1-expressing macrophages are located mainly in the peritumoral region of HCC using immunohistochemistry and flow cytometry. In other words, the presence of TAMs may play an important role in reducing the anticancer immune response of CD8^+^ and CD4^+^ T cells. Furthermore, we have demonstrated that inhibiting the expression of PD-L1 on macrophages may restore the function of CD8^+^ and CD4^+^ T cells. Therefore, targeting PD-L1-expressing macrophages in HCC may be used as a strategy to enhance the effectiveness of immunotherapy.

## Figures and Tables

**Figure 1 ijms-22-04710-f001:**
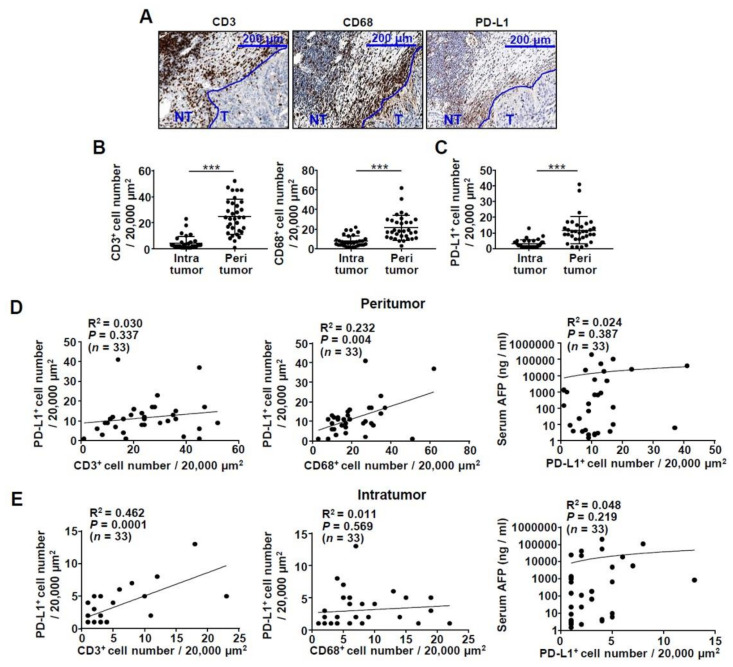
Patterns and correlations of CD3, CD68, and PD-L1-expressing cells in human HCC tissues: (**A**) a representative pattern of CD3, CD68, and PD-L1 expression in human tissues acquired through liver resection; (**B**,**C**) the number of CD3^+^ T cells, CD68^+^ macrophages, and PD-L1^+^ cells located in intratumoral and peritumoral region. *** *P* < 0.001; (**D**–**E**) correlation of CD3^+^ T cells, CD68^+^ macrophages, serum AFP, and PD-L1^+^ cells located in peritumoral and intratumoral region (*n* = 33). Abbreviations: AFP, alpha fetoprotein; HCC, hepatocellular carcinoma; PD-L1, programmed death ligand 1.

**Figure 2 ijms-22-04710-f002:**
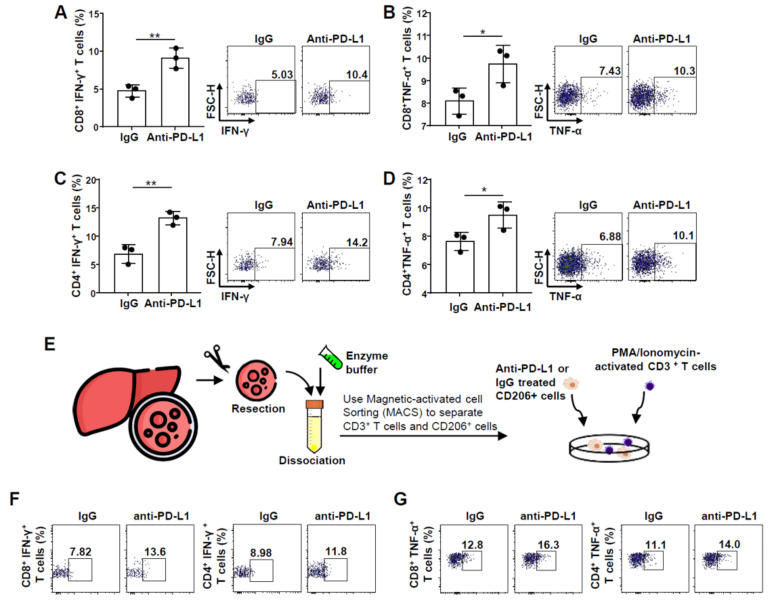
Functional enhancement of CD8^+^ and CD4^+^ T cells after co-culture with anti-PD-L1-treated macrophages: (**A**,**B**) expression and MFI of (**A**) IFN-γ, and (**B**) TNF-α, in CD8^+^ T cells when CD3^+^ T cells were co-cultured with PD-L1-blocked macrophages (*n* = 3) * *P* < 0.05, ** *P* < 0.01; (**C**,**D**) expression and MFI of (**C**) IFN-γ, and (**D**) TNF-α in CD4^+^ T cells when CD3^+^ T cells were co-cultured with PD-L1-blocked macrophages (*n* = 3) * *P* < 0.05, ** *P* < 0.01; (**E**) experiment schedule for separation of T cells and macrophages from the tissues acquired by hepatic resection; (**F**,**G**) differential expression of IFN-γ and TNF-α in CD8^+^ and CD4^+^ T cells when CD3^+^ T cells were co-cultured with PD-L1-blocked CD206^+^ macrophages and control cells from human tissues acquired through liver resection. Abbreviations: IFN-γ, interferon-γ; MFI, mean fluorescence intensity; PD-L1, programmed death ligand 1; TNF-α, tumor necrosis factor-α.

**Figure 3 ijms-22-04710-f003:**
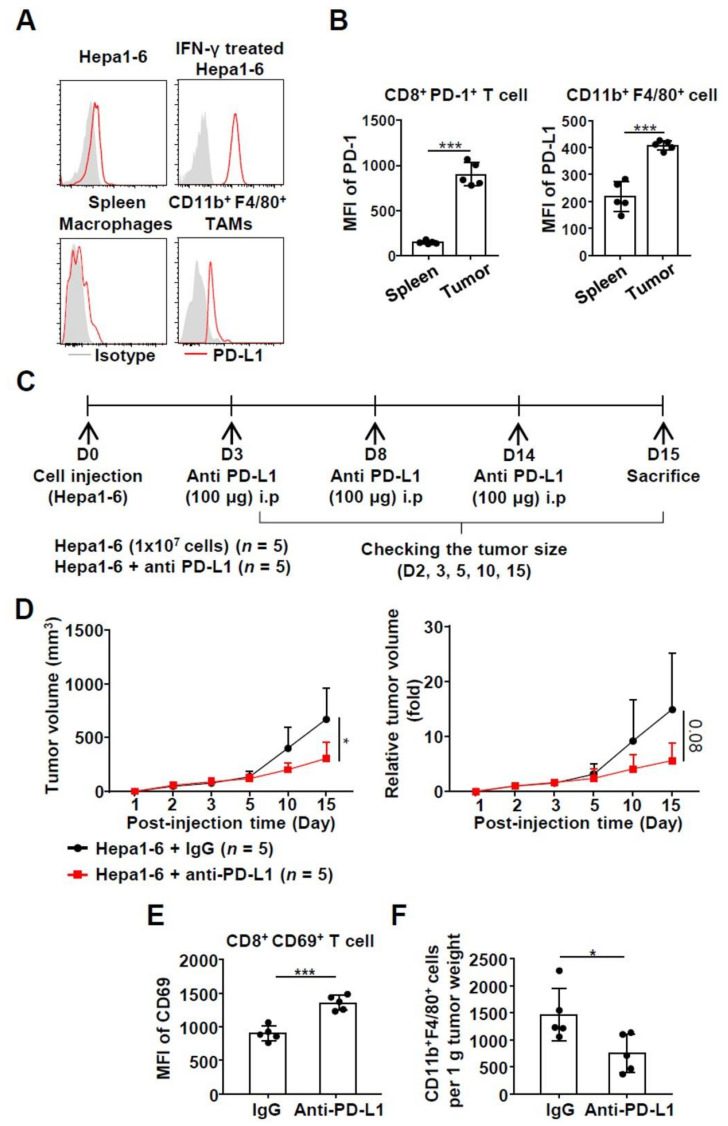
In vivo effect on intratumoral T cells and macrophages following anti-PD-L1 treatment in HCC syngeneic model; (**A**) expression of PD-L1 on non-treated Hepa1-6 cells, IFN-γ treated Hepa1-6, spleen macrophages, and CD11b^+^ F4/80^+^ macrophages; (**B**) PD-1 expression on CD8^+^ T cells in the spleen and tumor (left). PD-L1 expression on CD11b^+^ F4/80^+^ macrophages in the spleen and the tumor (right). (*n* = 5) ****P* < 0.001; (**C**) experimental schedule in the syngeneic HCC mouse model. In total, 100 µg of anti-PD-L1 antibody was intraperitoneally injected. (D0; Day0, D3; Day3, D8; Day8, D14; Day14, D15; Day15); (**D**) serial alteration in the size of the tumors generated from Hepa1-6 cells (1 × 10^7^ cells) after treatment with anti-PD-L1 antibody (absolute tumor volume; left, relative tumor volume; right). * *P* < 0.05; (**E**) increased activation of T cells when anti-PD-L1 on the CD11b^+^ F4/80^+^ macrophage surface was blocked. (*n* = 5). *** *P* < 0.001; (**F**) decreased the number of CD11b^+^ F4/80^+^ macrophages per 1g tumor weight when anti-PD-L1 on the CD11b^+^ F4/80^+^ macrophage surface was blocked. (*n* = 5). * *P* < 0.05. Abbreviations: HCC, hepatocellular carcinoma; IFN-γ, interferon-gamma; PD-1, programmed death-1; PD-L1, programmed death ligand 1.

**Figure 4 ijms-22-04710-f004:**
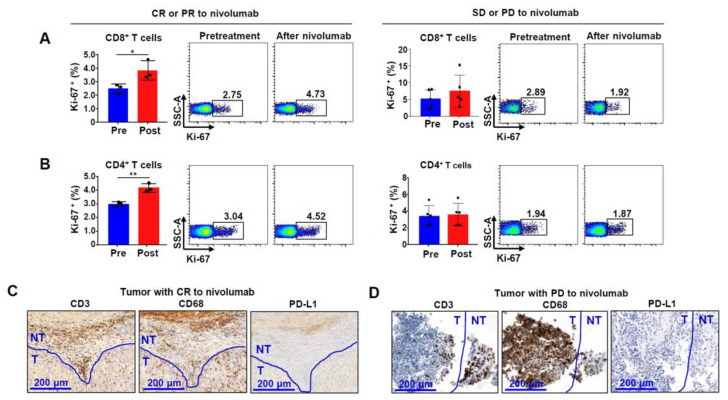
Ki-67 expression in peripheral CD8^+^ and CD4^+^ T cells before and after nivolumab treatment in patients with unresectable HCC: (**A**,**B**) differential expression of Ki-67 in CD8^+^ and CD4^+^ T cells from CR + PR (*n* = 3) and SD + PD (*n* = 5) groups. * *P* < 0.05, ** *P* < 0.01; (**C**,**D**) CD3, CD68, and PD-L1 expression patterns in patient tissues with CR (**C**) or PD (**D**) after nivolumab administration. Abbreviations: CR, complete response; HCC, hepatocellular carcinoma; PD, progressive disease; PD-L1, programmed death ligand 1; PR, partial response; SD, stable disease.

## Data Availability

The data presented in this study are available upon request from the corresponding author.

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
