# Peer review of "Preferential Expression of Programmed Death Ligand 1 Protein in Tumor-Associated Macrophages and Its Potential Role in Immunotherapy for Hepatocellular Carcinoma"

_ijms, 2021, doi:10.3390/ijms22094710_

Round 1
Reviewer 1 Report
The manuscript is well written and very easy to understand. Even if few experiments are showed, in my opinion they are well designed and show interesting results, even if the human part is a bit poor.
I would only suggest to better describe the experiments showed in section 2.3, as the sentence:
"CD8+ T cells and CD11b+ F4/80+ macrophages were acquired from the tumor and spleen, respectively"
is not very clear. At first read it seems that authors acquired CD8 cells from the tumor and CD11b cells from the spleen, while in the result they show that they isolated cells from both tissues.
Author Response
- The manuscript is well written and very easy to understand. Even if few experiments are showed, in my opinion they are well designed and show interesting results, even if the human part is a bit poor.
I would only suggest to better describe the experiments showed in section 2.3, as the sentence:
"CD8+ T cells and CD11b+ F4/80+ macrophages were acquired from the tumor and spleen, respectively"
is not very clear. At first read it seems that authors acquired CD8 cells from the tumor and CD11b cells from the spleen, while in the result they show that they isolated cells from both tissues.
Thank you for your suggestion. We have corrected the sentence in the revised manuscript. (page 4, line 22)
Reviewer 2 Report
Preferential Expression of Programmed Death-Ligand 1 Protein in Tumor-Associated Macrophages and Its Potential Role in Immunotherapy for Hepatocellular Carcinoma
Comments to the Editor/Authors:
- Since the article heavily focuses on molecular (immune)biology in respect to HCC, the research within the manuscript clearly fits within the scope of IJMS as well as that of Cancers.
- Summary: As no link between the expression of PD-L1 in tumor-associated macrophages and a response to anti-PD-1 treatment has been established for HCC, the authors sought to investigate whether a correlation between the two does exist. The authors did so for the purposes of determining whether PD-L1 expression and its effects on other cells within the TME could both serve as a biomarker and be quelled to both reinstate the natural immune response and aid in the efficacy of immunotherapies (i.e., checkpoint inhibitors). PD-L1 expression was determined from cellular and ex vivo studies as well as from patient specimens (both treated and untreated with anti-PD-L1 treatment).
- The experiments that the authors performed are appropriate for the intended purposes.
- The article’s conclusions of the results from the authors experiments are appropriate.
- In the Results section 2.1, there appears to be essentially a redundant statement that I believe could be removed. The sentence, “CD68+ macrophages were mainly located in the peritumoral region than in the intratumoral region.”, can be readily inferred from the preceding statement and appears unnecessary to state. That is, the finding that there exists a higher number of CD68+ cells in the peritumoral region compared to the intratumoral region allows a reader to assume that CD68+ macrophages primarily reside there.
- For the plots of Figure 1D & 1E, I believe the adjusted R2 (coefficient of determination) would be more appropriate (than r) due to it accounting for the population/sample size. As a minor revision, could you please insert the Adjusted R2 values for those plots?
Author Response
- Since the article heavily focuses on molecular (immune)biology in respect to HCC, the research within the manuscript clearly fits within the scope of IJMS as well as that of Cancers.
- Summary: As no link between the expression of PD-L1 in tumor-associated macrophages and a response to anti-PD-1 treatment has been established for HCC, the authors sought to investigate whether a correlation between the two does exist. The authors did so for the purposes of determining whether PD-L1 expression and its effects on other cells within the TME could both serve as a biomarker and be quelled to both reinstate the natural immune response and aid in the efficacy of immunotherapies (i.e., checkpoint inhibitors). PD-L1 expression was determined from cellular and ex vivo studies as well as from patient specimens (both treated and untreated with anti-PD-L1 treatment).
- The experiments that the authors performed are appropriate for the intended purposes.
- The article’s conclusions of the results from the authors experiments are appropriate.
Thank you for the careful review.
- In the Results section 2.1, there appears to be essentially a redundant statement that I believe could be removed. The sentence, “CD68+ macrophages were mainly located in the peritumoral region than in the intratumoral region.”, can be readily inferred from the preceding statement and appears unnecessary to state. That is, the finding that there exists a higher number of CD68+ cells in the peritumoral region compared to the intratumoral region allows a reader to assume that CD68+ macrophages primarily reside there.
Thank you for your comment. As suggested, we have deleted the
sentence in the revised manuscript.
- For the plots of Figure 1D & 1E, I believe the adjusted R2 (coefficient of determination) would be more appropriate (than r) due to it accounting for the population/sample size. As a minor revision, could you please insert the Adjusted R2 values for those plots?
Thank you for your careful review of our manuscript. We have
replaced R values with R2 values in the revised Figure 1D and 1E.